# The Pivotal Role of Oxidative Stress in the Pathophysiology of Cardiovascular-Renal Remodeling in Kidney Disease

**DOI:** 10.3390/antiox10071041

**Published:** 2021-06-29

**Authors:** Verdiana Ravarotto, Giovanni Bertoldi, Georgie Innico, Laura Gobbi, Lorenzo A. Calò

**Affiliations:** Nephrology, Dialysis and Transplantation Unit, Department of Medicine, University of Padova, 35128 Padova, Italy; verdiana.ravarotto@gmail.com (V.R.); giovanni.bertoldi@studenti.unipd.it (G.B.); georgie.innico@unipd.it (G.I.); laura.gobbi_01@aopd.veneto.it (L.G.)

**Keywords:** oxidative stress, renin-angiotensin system, cardiovascular disease, chronic kidney disease, dialysis

## Abstract

The excessive activation of the renin-angiotensin system in kidney disease leads to alteration of intracellular pathways which concur altogether to the induction of cardiovascular and renal remodeling, exposing these patients since the very beginning of the renal injury to chronic kidney disease and progression to end stage renal disease, a very harmful and life threatening clinical condition. Oxidative stress plays a pivotal role in the pathophysiology of renal injury and cardiovascular-renal remodeling, the long-term consequence of its effect. This review will examine the role of oxidative stress in the most significant pathways involved in cardiovascular and renal remodeling with a focus on the detrimental effects of oxidative stress-mediated renal abnormalities on the progression of the disease and of its complications. Food for thoughts on possible therapeutic target are proposed on the basis of experimental evidences.

## 1. Introduction

Oxidative stress arises when the physiologically-produced oxidant species overwhelm the endogenous antioxidant recovery capacity of the cells, leading to loss of intracellular redox homeostasis [1]. As net result of this alteration, reactive oxygen species (ROS) and reactive nitrogen species (RNS) affect and damage lipids, DNA, proteins, tissues and organs [2].

The highly complex renal pathways involved in the induction of oxidative stress entail the activation of the renin-angiotensin system (RAS). This latter in fact, by regulating several components of the cardiovascular-renal system, influences also pathophysiological responses through hypertension, rapid progression of atherosclerosis, and vascular remodeling [3]. Particularly in kidney disease, the excessive activation of the RAS and the excessive presence of oxidative stress participate together as playing a game in the progression of structural changes in the vessels and in the heart. Both oxidative stress and RAS are very active players in a fight between endogenous antioxidant defenses, oxidative stress and its mediators where progressive renal functional decline remains the natural most frequent result if no approach both pharmacologic and nutritional is started (Figure 1).

This review will explore the current knowledge on the RAS and the influence of oxidative stress in kidney disease with a focus on patients with chronic kidney disease stage 3–4 (CKD) and kidney failure with or without replacement therapy and on the possible available supportive therapeutic approach for these patients.

## 2. The Renin-Angiotensin System in Chronic Kidney Disease

Ang II signaling plays a pivotal role in the vasoconstriction of renal vascular smooth muscle cells (VSMCs) and in the sodium and water handling, involved in the pathophysiology of renal and cardiovascular remodeling. [4]. The signaling cascade triggered by Ang II depends on its membranes receptors subtype-1 (AT1R) which promote vasoconstriction, inflammation, fibrosis, cellular growth, or its receptors subtype-2 (AT2R) mediating vasodilation, insulin sensitivity, anti-remodeling, anti-atherogenesis effects. High levels of Ang II are a hallmark for kidney diseases. 

Chronic Ang II stimulation induces elevation of blood pressure (BP) due to juxtaglomerular injury, hypoxia, tubular necrosis and interstitial fibrosis. Experimental evidence show in fact, that renal autoregulation is relatively ineffective in subjects with CKD, in hypertensive and in diabetic patients as afferent arteriolar vasoconstriction is not able to counteract the increased systemic BP and to prevent its spreading to the glomeruli [5]. These evidences further support the rationale, already reported in several clinical trials, of using ACE inhibitors (ACEi) and angiotensin receptor blockers (ARBs) as antihypertensives with pleiotropic effects which may be translated into cardiovascular and renal protection [6].

Overall, the RAS and, in particular, its major effector molecule Ang II, are principally involved in cardiovascular-renal remodeling in CKD driving oxidative stress, inflammation and immune responses (Figure 1).

## 3. Sources of Oxidative Stress

Oxidative stress is related to an imbalance in reactive oxygen species (ROS) that are generated during the sequential reduction of molecular oxygen. ROS usually refers to free radicals and other non-radical intermediates having the capacity to quickly react with all the surrounding molecules. The most representative reactive specie is superoxide (O_2_^•−^), produced by the respiratory chain in the mitochondria and by several enzymes such as nicotinamide adenine dinucleotide phosphate (NAD(P)H) oxidase, uncoupled nitric oxide synthase (NOS), xanthine oxidase, cytochromes P450, lipoxygenase (LOX), cyclooxygenase (COX).

Superoxide, has a very short half-life and might immediately or catalytically dismutate to hydrogen peroxide (H_2_O_2_), a ROS with a greater stability able to cross membranes, which also undergoes catalytic decomposition to ^•^OH, and acts as a signaling molecule together with superoxide itself. Further, superoxide favors the production of others extremely reactive and toxic compound such as radical nitric oxide (NO^•^) and peroxinitrites (ONOO^−^) [7].

Among the enzymes generating O_2_^•−^, xanthine oxidoreductase is the rate-limiting enzyme that catalyzes the conversion of hypoxanthine to xanthine and subsequently of xanthine to uric acid. It entails the dehydrogenase form and the oxidase form that is specifically involved in pro-inflammatory status by providing electrons to molecular oxygen with ensuing production of O_2_^•−^ and H_2_O_2_ [8,9]. The NAD(P)H oxidases (Noxs) family comprises several enzymes that catalyze the transfer of electrons from NAD(P)H to two flavin elements and heme moieties to O_2_. These enzymes are present with seven isoforms that can be found in cardiovascular, renal tissues and thyroid cells: Nox1, Nox2, Nox3, Nox4, Nox5, Duox1 and Duox2 [10]. They encompass several cytoplasmic and transmembrane proteins that altogether set up the catalytic subunit. Nox2 (gp91^phox^) is the ubiquitarian NAD(P)H oxidase found in phagocytes and present in the cardiovascular system. It requires the cytoplasmic p47^phox^, p67^phox^, p40^phox^ subunits, the transmembrane p22^phox^ and the small G protein Rac1/2 to create the active cytochrome b_558_ [11]. Diversely from constitutively active Noxs, that generate O_2_^•−^ to influence transcription factors and intracellular signaling pathway of growth, inflammation and contraction, Nox2 triggers ROS production only upon adequate stimulation [12,13].

Noxs are abundantly expressed in cardiomyocytes, cardiac cells, endothelial cells and VSMCs and take a role in several pathological conditions due to their interconnection with the RAS: Ang II via AT1R activates Nox2, increases ROS production and triggers other intracellular pathways such as the extracellular regulated signal kinase ERK 1/2 and the RhoA/Rho kinase (ROCK) 1/2 involved in profibrotic responses and cardiovascular-renal remodeling [14,15].

Intracellular redox state is also strictly linked to the presence of iron and other metals (copper, chromium, cobalt, vanadium) physiologically balancing between redox and oxidized form. Increased superoxide promotes excessive release of Fe^2+^ from cellular storage rendering it available for Haber-Weiss and Fenton reaction with ensuing production of other ROS. This is particularly evident and harmful in dialysis patients, where blood cells are extremely stressed and the toxic trace ions tend to accumulate and to worsen with the duration of the dialysis [16].

## 4. Oxidative Stress Damage in CKD and Kidney Failure with or without Replacement Therapy Patients

Oxidative stress in patients is a well-known feature of CKD and kidney failure with or without replacement therapy. The molecular mechanisms underlying oxidative damage are multiple and strictly intertwined in a vicious circle to favor the progression of kidney disease (Figure 1 and Table 1) [17]. 

As an example, permanent exposure of red blood cells to high oxygen concentration favors the production of ROS and the reduction of antioxidant defenses with ensuing oxidative injury to erythrocytes [18]. The erythrocytes’ plasma membrane displays in fact, the conjugate-protein hemoglobin which is deputed to binding and release of oxygen and carbon dioxide making erythrocytes more susceptible to oxidative damage [19]. Moreover, the absence of cell organelles and nucleus renders erythrocytes unable to repair their damaged components (such as the membrane-SH groups) thereby, the presence of oxidative damage favors erythrocytes aging and significantly affects the viscoelastic properties of their membrane [19]. Likewise, lipid peroxidation affects cellular membranes producing malonaldehyde (MDA), an end product recognized to have a role in the onset of atherosclerosis [20]. As an example, levels of MDA are significantly increased in patients with Fabry disease, an X-linked storage disorder characterized by deficient expression of α-GalA enzyme. Fabry disease progresses to severe renal and cardiovascular manifestations due to the massively impact of oxidative stress, which negatively influence the prognosis [21]. In addition, oxidative stress is linked to increased levels of indoxyl sulfate, a membrane-bound toxin which contributes to drive phosphatidylserine to the external membrane in peripheral mononuclear cells and in erythrocytes showing activation of apoptotic processes [22]. These biochemical characteristics are particularly relevant for those patients undergoing dialysis treatment where blood components are heavily exposed to mechanical forces and biochemical rearrangement. 

In kidney disease many other cellular components are deeply impaired by ROS action in an irreversible fashion due to a “positive feedback” that spreads the damage to the surrounding structures and for the concomitant reduction of antioxidant defenses. 

A boost of mitochondrial respiratory chain is responsible for overexpressing p22^phox^ in NOXs (with ensuing production of superoxide), but also for activation of other enzymes such as xanthine oxidase, lipoxygenase, cyclooxygenase and P450 monooxygenase and for release of ROS from endoplasmic reticulum and nuclear envelope [23]. Besides, antioxidant mechanisms such as the antioxidant nitric oxide (NO) release, the activity of superoxide dismutase (SOD1, SOD2) and that of glutathione peroxidase (Gpx1) are concomitantly downregulated. Mitochondrial dysfunction is also associated to insulin resistance in the kidney for the diminished capacity of tissues and cells to respond to insulin levels. This process is related to the increase and accumulation of circulating free fatty acids (FFAs) and triglycerides and to increased susceptibility of low density lipoproteins (LDL) to oxidative modifications [24]. All these pathways are deeply involved in cardiovascular and renal remodeling and concur to the progression of long-term hypertensive complications such as left ventricular hypertrophy [25]. We observed that in patients undergoing dialysis presenting left ventricular hypertrophy, both p22^phox^ and oxidized LDL (oxLDL) levels correlated with the left ventricular mass [26]. These evidences further support the rationale of reducing oxidative stress as therapeutic option in order to decrease the expression of proteins and markers that are firmly correlated with cardiovascular disease in patients with CKD, kidney failure with or without replacement therapy. In this regard it would be useful to exploit, besides pharmacological interventions, nutritional approaches such as integration with polyphenols and ω-3 fatty acids (eicosapentaenoic acid-EPA and docosapentaenoic acid-DPA) aimed at reducing oxidative stress and inflammation, which could further provide cardiovascular and renal protection [27]. Interestingly in our dialyzed patients, the supplement with green tea reduced significantly p22^phox^ levels and oxLDL with concomitant significant reduction of the left ventricular mass providing clinical evidence for this hypothesis [26]. 

The presence of oxidative stress influences the transcription of several factors involved in proliferation, fibrosis and atherosclerotic events as well. Plasminogen activator inhibitor (PAI)-1 is the inhibitor of both tissue plasminogen activator and urokinase, involved in fibrinolysis, and favors hypercoagulable state in diabetes patients [28]. In addition, in CKD PAI-1 plays a central role in the progression of nephropathy via glomerulosclerosis and interstitial fibrosis induction [29]. It was originally established that two potent inducers of PAI-1 are TGFβ and the RAS. Deep investigations elucidated that the intracellular pathways responsible for PAI-1 induction are the mitogen activated kinases (MAPK), ROCK, NF-κB, AP1 (activator protein -1, a c-Jun homodimer) and SP1 (a general transcription factor) all of them induced by oxidative stress [30,31,32,33]. 

Oxidative stress in kidney failure with or without replacement therapy is present since the very early stages of the renal dysfunction, as it is demonstrated by the presence of 8-iso-PGF(2alpha) and C-reactive protein [34]. As renal disease progresses and patients undergo renal replacement therapy with dialysis treatment, endothelial dysfunction, measurable in terms of flow-mediated dilation (FMD), increases as well as atherosclerotic remodeling, observable in terms of increased intima-media thickness (IMT). In addition, both FMD and IMT are correlated with C-reactive protein [35]. These biochemical mechanisms responsible for renal remodeling are triggered by oxidative stress and hormonal stimuli such as Ang II, endothelin (ET)-1 and TGFβ particularly in the glomeruli where the activation of the transcription factor yes-associated protein (YAP) mediates the endothelial-to-mesenchymal transition (EMT) of endothelial cells [36,37]. This phenotypic transition implies the loss of cell junctions’ components as E-cadherin in favor of smooth muscle markers as α smooth muscle actin (αSMA) and lead to cell motility and to a cellular population producing αSMA and matrix proteins, including collagen. Among kidney diseases, the hypertensive nephropathy is the most representative example of these mechanisms. In hypertensive nephropathy the continuous stress due to high blood pressure in the afferent arterioles induces podocytes injury with loss of their selective barrier, increased media thickness due to hyalinosis and glomerular tuft shrinkage [37]. Tubular injury do to hypertension is associated to damage of peritubular capillaries which triggers hypoxia, microvasculature dysfunction, inflammation and finally turns into intersitial fibrosis [37]. Together with chronic uncontrolled hypertension, BP variability is also associated to a high risk of renal outcomes [38,39]. Observations from our cohort of CKD patients linked BP variability with a prevalence of sleep disorders such as obstructive sleep apnea, insomnia and restless leg syndrome further supporting the hypothesis that endothelial dysfunction and glomerular damage play a crucial role in the development of BP variation and in worsen disease progression in these patients [40,41,42]. Evidences for the correlation of oxidative stress and BP variability can be observed in population at high cardiovascular risk such as subjects with familial history of essential hypertension and myocardial infarction and type 2 diabetes mellitus and hypertension where BP variations significantly correlate with 8-hydroxydeoxyguanosine, 8-isoprostane and diacron-reactive oxygen metabolites respectively [43,44]. 

The endogenous antioxidant responses against oxidative stress are multiple and involve the intervention of low molecular weight compounds such as NO, scavenger enzymatic mechanisms in terms of superoxide dismutases (SODs), glutathione peroxidases (GSH-Px) and more complex cellular responses such as autophagy that, depending on the entity of oxidative stress, can be either protective or harmful [45]. A well-known mechanism observed in CKD and dialysis patients is the induction of the inducible heme oxygenase isoform-1 (HO-1) literally induced by oxidative stress. This enzyme acts on heme moieties to produce biliverdin, subsequently metabolized to bilirubin—a powerful antioxidant itself, and to produce the vasodilator CO. The release of bilirubin and CO may support maintain vascular tone, blood pressure and endothelial function properly regulated. Moreover, the chronic induction of HO-1 might provide myocardial protective effects in kidney failure with or without replacement therapy in the long term [46]. Evidences in dialysis patients in which were used vitamin E- coated membranes as dialyzers or treated with green tea, showed increased level of HO-1 confirming once again, the importance of recurring to antioxidants strategies in this population at high cardiovascular risk [26,47].

Oxidative stress plays an harmful role also in patients receiving kidney transplant and it is strictly connected to post transplant hypertension [48,49,50]. Patients who undergo transplantation often develop hypertension during treatment with calcineurin inhibitors (CNIs). In a cohort of post- renal transplant hypertensive patients from our Unit, we observed a significant increase of p22^phox^ expression in addition to reduced antioxidant defenses such as HO-1 and total plasma antioxidant power when compared to normotensive transplanted recipients [51]. Interestingly, in those hypertensive patients, a counterregulatory mechanism is also activated regardless CNIs-induced vasoconstriction and hypertension: NO synthase and NO metabolites plasma levels are increased together with increased hydroperoxides and peroxinitrites showing an upregulation of the NO system [52]. The concomitant NO system upregulation and CNIs-induced vasoconstriction might seem a contradiction, yet it could be explained by the high presence of ROS driven by CNIs, which reduce NO bioavailability and its vasodilatory action by reacting with it to produce peroxinitrites. A further concomitant mechanism promoting post-transplant hypertension is the excessive sodium retention driven by Ang II, the Rho A/ROCK system and oxidative stress itself, likely through the modulation of the sodium chloride cotransporter in the distal convoluted tubule [48]. Taken together, these evidences further support the rationale of using in kidney transplant recipients, antihypertensive drugs with demonstrated effects on reduction of oxidative stress [48,50].

## 5. Cardiovascular Risk Factors in CKD and in Kidney Failure with or without Replacement Therapy

Traditional risk factors such as smoke, diabetes and hypertension do not entirely justify the high incidence of cardiovascular morbidity and mortality observed in patients with CKD and kidney failure with or without replacement therapy. Other so-called “non- traditional” risk factors such as inflammation, endothelial dysfunction and in particular increased oxidative stress have been recognized as playing a major role in the induction of cardiovascular remodeling [26,53,54].

Left ventricular hypertrophy (LVH) is the most common myocardial structural abnormality associated with chronic kidney diseases [55]. Among the intracellular pathways involved in the stimulation of fibrotic responses, Rho A/ROCK signaling cascade plays a pivotal role by further mediating upregulation of ROS through the induction of Nox (Figure 2) [56,57]. The binding of Ang II with its AT1R induces via G protein signaling high expression of p63RhoGEF of p115RhoGEF, inducers of Rho A/ROCK signaling cascade. RhoA/ROCK activation leads to cell proliferation and cardiovascular-renal remodeling [58,59]. A study from our laboratory showed that the expression of p63RhoGEF was significantly increased in hypertensive subjects compared to normotensive healthy individuals and that correlated with both systolic BP and diastolic BP matching the increased expression of the myosin phosphatase target protein (MYPT)-1 [60]. MYPT-1 is the target of ROCK phosphorylation and is the regulatory subunit of myosin light chain phosphatase (MLCP). The phosphorylation of MYPT-1 in fact, by inhibiting MYPT-1 phosphatase activity, increases MLC kinase activity, promotes VSMC contraction and Ca^2+^ handling [61] and induction of genes involved in inflammation, oxidative stress and oxidative stress mediated fibrosis processes. Compared to stage 3–4 CKD patients and patients undergoing dialysis without LVH, those who display LVH have also a remarkable increased phosphorylation of MYPT-1 which positively correlates to LV mass [62]. The incubation of leukocytes isolated from stage 3–4 CKD and dialysis patients with the ROCK inhibitor fasudil significantly dose-dependently decrease MYPT-1 phosphorylation, suggesting that regulation of ROCK might be useful to prevent cardiovascular-renal remodeling [62]. In addition, an upstream block of Ang II via type 1 Ang II receptor blockers, is able to reduce significantly p63RhoGEF and MYPT-1 phosphorylation further confirming that any attempt to reduce ROCK activity is crucial for the progression of cardiovascular-renal remodeling and that patients at high cardiovascular risk such as hypertensive, CKD patients might benefit from the pleiotropic effects of angiotensin receptor blockers [63].

The activation of ROCK tracks not only with vascular contraction but also with the expression of proinflammatory cytokines and molecules of adhesion, with the expression of atherothrombogenic PAI-1, and with the regulation of cyclophilin A [64]. Through its direct downstream target, ezrin, radixin, moesin (ERM) and LIM kinase in fact, ROCK influences cytoskeletal rearrangement and ERK 1/2 and other mitogen activated protein kinases (MAPK) through oxidative stress [65]. The activation of ERK 1/2 is pivotal for integrated responses that activates transcription factors (e.g., c-myc, c-jun and ATF2), chromatin phosphorylation and for the balance of eccentric/concentric growth of the heart [66,67]. The treatment of patients undergoing hemodialysis with the antioxidant green tea reduced significantly the active phosphorylated state of ERK 1/2 and was associated to reduced left ventricular mass further supporting a role of ERK 1/2 in the oxidative stress-mediated profibrotic responses [26]. 

Among the implications of increased left ventricular mass, atrial fibrillation is the most frequent arrhythmia in patients with kidney failure with or without renal replacement therapy and is associated to marked MYPT-1 phosphorylation [68]. Cardiac remodeling, including atrial myocytes structural and electrical alterations, greatly contributes to the pathogenesis of atrial fibrillation. In this context, Connexins (Cxs), integral membrane proteins components of the gap junctions in heart cells, are necessary for the rapid cell-to-cell transfer of action potential and alterations in Cxs expression are just a hallmark of atrial fibrillation. Evidences for this relation have been found in animal models of chronic kidney disease, where increased susceptibility to atrial fibrillation correspond to Cxs altered expression (in particular Cx40 and Cx43) and is a consequence of Ang II signaling and RhoA/ROCK pathway [69]. Our recent study in dialysis patients demonstrated that patients with atrial fibrillation, contrarily to those dialysis patients with no atrial fibrillation, had increased ROCK activity in terms of MYPT-1 phosphorylation which correlates positively with increased Cx40 level. In addition, the incubation of peripheral mononuclear cells of these patients with the ROCK inhibitor fasudil reduced Cx40 protein expression supporting the hypothesis that Cx40 may be a downstream target of RhoA/ROCK pathway [70]. These results add therefore additional evidences for the involvement of RhoA/ROCK pathway in the perpetuation of atrial fibrillation, also in the light of the role of MAPK in theregulation of Cxs expression [71]. 

Alongside specific pathways inducing fibrosis and electrical remodeling, CKD and dialysis patients exhibit also reduced bioavailability of antioxidant defenses such as NO which further influence the dysfunctional VSMCs migration toward intima. This migration causes intimal hyperplasia alongside with abnormal extracellular matrix deposition and hyaline material accumulation, vascular calcification with 0arteries stiffening and high pulse pressure [35,72]. The balance between NO, ROCK and ERK 1/2 pathways finally converges also in the fine-tuning of HO-1 [73,74]. 

In a biochemical framework of increased oxidative stress and decreased endogenous antioxidant defenses, altered intracellular pathways leading to atherosclerosis, fibrosis and more in general to cardiac and renal remodeling, an efficient therapeutic measure could be the reduction of oxidative stress with pharmacological or nutritional options. Many clinical trials have been conducted targeting oxidative stress in attempt to reduce cardiovascular incidence and oxidative stress-related sequelae, however, most of them have failed to prove a significant clinical benefit in reduction of atherosclerosis and its complications [75]. This may be due to the selection of inappropriate antioxidant drugs, or their combinations, or even the dosage, the population included or duration of the trials and their follow ups [76]. Nevertheless, oxidative stress still undoubtedly maintains a pivotal role in the pathophysiology of cardiovascular and renal remodeling, therefore its reduction is one of the most important goal that has to be achieved, by focusing on right protocols, right molecules and biochemical pathways. As an example, there is a growing interest onSLGT2 inhibitors as antioxidant alternatives to vitamins or natural polyphenols extracts. Those drugs are currently used in type 2 diabetes mellitus as oral hypoglycemic treatment however, they also have a direct effect on the reduction of Nox4 expression, in the generation of free radicals and on endothelial NOS (eNOS) and xanthine oxidase expression and activity [77]. Moreover, they have been reported as pleiotropic agents for their ability to reduce advanced glycosylation end products (AGEs) and proinflammatory cytokines and to improve mitochondrial dysfunction, hemodynamic state and to reduce RAS activity [77].In addition, the glucagon-like peptide 1 (GLP-1), an hormone that stimulates insulin secretion and downregulates glucagon secretion [78] has been shown to decrease oxidative stress and to promote antioxidant effects through the activation of nuclear erythroid-2 like factor-2 (Nrf2) and to improve mitochondrial function [79], These properties have represented the rationale for molecules such as GLT-1R agonists for their use in type 2 diabetes where they also are credited for a renal-cardiovascular protection.

## 6. Oxidative Stress and Dialysis Procedures

Hemodialysis and hemofiltration are the main renal replacement therapies for subjects with end-stage renal diseases. Failure of kidney function leads to uremia and accumulation in the bloodstream of toxic uremic compounds such as low-molecular weight solutes (urea, phosphorus, and creatinine), middle molecules (β_2_-microglobulin and fibroblast growth factor 23), parathyroid hormone, protein-bound solutes, AGEs, and trimethylamine N-oxide (TMAO) [80]. Both extracorporeal and peritoneal dialysis entail the use of a dialytic filter and a dialysis solution which permit the release of toxic molecules from blood and the concurrent intake of essential solutes. In peritoneal dialysis the peritoneum itself is exploited as the natural semipermeable membrane, while, in hemodialysis a manufactured membrane is the core of the dialytic filter.

It is evident that during those renal replacement therapies blood cells are extremely forced against the membrane walls and against dialyzer/filter components. By leaving the vessels, blood loses the protective effect of the endothelium and by coming into contact with synthetic surfaces and by changing the geometry of the flow blood’s components activate leukocytes and increase oxidative stress [81]. Activation of phagocytic cells induces a respiratory burst with a rapid release of O_2_^•−^ and H_2_O_2_ spreading high reactivity to surrounding molecules, [82]. Further, particularly in hemodialysis patients, oxidative stress shortens lifetime of erythrocytes by up to 70% leading to anemia and to accumulation of toxic compounds in the blood [83,84].

Regarding extracorporeal dialysis, several types of membranes are available for dialyzers, spanning from high or low flux to high or low permeability and functionalized as needed [47,85]. Being the internal layer the very first interaction with blood before separation, functionalized membranes are of particular interest. Their physicochemical characteristics are responsible for biochemical cascades as coagulation, platelet adhesion and protein adsorption with net effects on solute removal [86]. In this regard, vitamin E coated dialyzers are adopted with the aim to further improve biocompatibility and to provide a antioxidant protection to circulating blood cells and lipoproteins [87,88]. Good evidence of this may come from a study in our cohort of patients with kidney failure undergoing dialysis with a vitamin E-coated dialyzer, where we observed that one year of treatment decreased significantly the expression of oxidative stress and inflammation proteins and related markers associated with cardiovascular disease [47]. In particular, p22^phox^ was significantly decreased already after six months of treatment and further declined after twelve months demonstrating a reduction of oxidative stress and blockade of leukocytes’ activation. In parallel, LDL oxidation and expression of PAI-1 were reduced as well as the phosphorylation state of ERK 1/2. In addition, the coating with vitamin E increased HO-1 expression further supporting an active role of vitamin E in the induction of antioxidant intracellular pathways through Nrf2 and Akt [47]. 

Another extracorporeal renal replacement procedure proven to be effective against oxidative stress is hemodiafiltration with online regeneration of ultrafiltrate (HFR). A double-chamber hemodialysis filter permit reinfusion of regenerated ultrafiltrate through a charcoal-resin cartridge. Due to its particular structure, the system combines diffusion, convection and absorbance: the ultrafiltrate is processed in a charcoal-resin component of the cartridge which has the peculiar characteristic to absorb pro-inflammatory cytokines. Subsequently, the regenerated ultrafiltrate is reinfused into the bloodstream before the diffusive section of the filter [85]. This technique has been reported to reduce IL-6, tumor necrosis factor (TNF)-α, C-reactive protein and, further, compared to the standard bicarbonate dialysis, we showed to be effective on oxidative stress in terms of reduction of p22^phox^ gene and protein expression and of reduction of PAI-1 as well as LDL oxidation. In addition, also HFR promotes a significant increase of gene and protein expression of HO-1 [85,89].

In peritoneal dialysis, oxidative stress has been reported only in terms of increased AGEs, and other pro-oxidants derived by glucose [90]. Very recently, by using a molecular biology approach, we have observed in peritoneal dialysis patients that, oxidative stress is increased by peritoneal dialysis procedure as shown by the significantly increased levels of oxidative stress markers such as p22^phox^, ROCK activity (MYPT1) and ferritin levels which all further increased after 6 months [91], therefore the use of more biocompatible dialysis solutions containing different glucose polymers such as icodestrin should be considered for their possible beneficial effects on oxidative stress and inflammation ( ).

## 7. Conclusions

The kidney is extremely vulnerable to damage caused by oxidative stress. The progression of kidney disease engenders biochemical mechanisms that alter cellular homeostasis and vascular structures, which in the long terms results in cardiovascular and renal remodeling with deleterious outcomes. It is now clear that oxidative stress plays a very important role in both the onset and progression of kidney diseases. Targeting oxidative stress and the related molecular mechanisms together with the availability of specific dialytic procedures which have provided high efficacy against oxidative stress are therefore a necessary objective to improve the prognosis of CKD and dialysis patients.

## Figures and Tables

**Figure 1 antioxidants-10-01041-f001:**
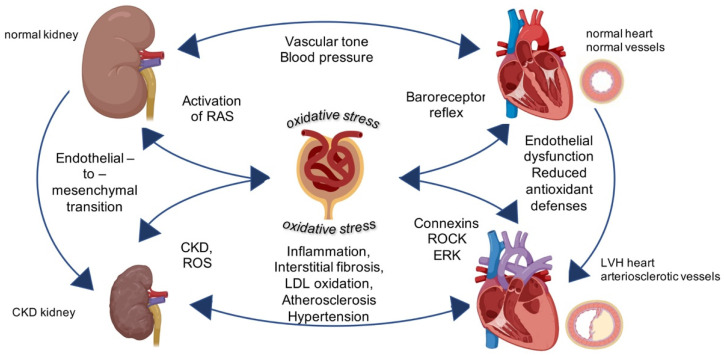
Central role of oxidative stress for cardiovascular-renal remodeling in kidney diseases. Physiological activity of kidney regulates vascular tone and controls blood pressure. The intracellular mechanisms triggered by oxidative stress targeting the kidney induce chronic kidney diseases and left ventricular hypertrophy (LVH) in the vessels and in the heart leading to general cardiovascular-renal remodeling. Image created with BioRender.com.

**Figure 2 antioxidants-10-01041-f002:**
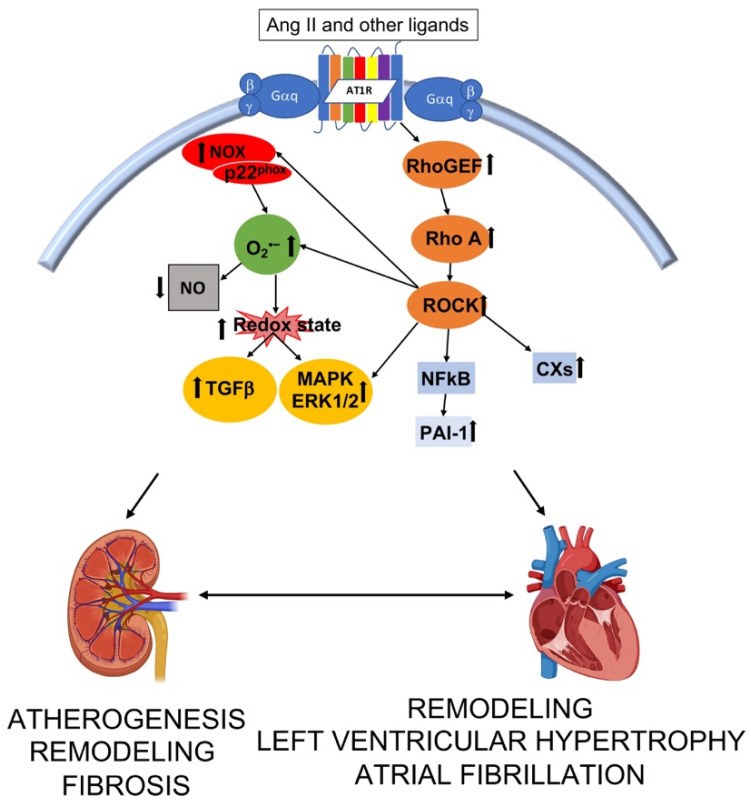
Interplay of intracellular pathways leading to structural alteration and function in kidney and heart. Ang II—angiotensin II; RhoGEF—Rho guanosine exchange factor; ROCK –Rho kinase; NFκB—nuclear factor kappa B; PAI-1—plasminogen activator inhibitor 1; CXs—connexins; NOX—nicotinamide adenine dinucleotide phosphate oxidases; O_2_^•−^—superoxide; NO—nitric oxidie; TGFβ —transforming growth factor β; MAPK—mitogen activated protein kinase; ERK 1/2—extracellular regulated signal kinase. Image created with BioRender.com.

**Table 1 antioxidants-10-01041-t001:** Schematic summary of molecular mechanism involved in the cardiovascular-renal remodeling. To the molecular mechanisms taking part to the cardiovascular-renal remodeling listed in the table, correspond the principal effectors involved and the relative compartment where the mechanisms take place. ANG II—angiotensin II; PAI 1—plasminogen activator inhibitor 1; YAP—yes associated protein; RhoA/ROCK—Rho kinase; MAPK—mitogen activated protein kinase; oxLDL—oxidized low density lipoprotein.

Molecular Mechanism	Principal Effectors	Compartment
**ROS production**	NAD(P)H oxidase [11], Xanthine oxidase	cardiomyocytes endothelial cells VSMCs
**Interstitial fibrosis**	Ang II PAI 1	glomeruli tubular cells
**EMT**	YAP Ang II	glomeruli tubular cells
**Atrial fibrillation**	Connexins RhoA/ROCK MAPK	cardiomyocytes
**Endothelial dysfunction**	uncoupled nitric oxide synthase NAD(P)H oxidase ROCK 8-iso-PGF(2alpha) C-reactive protein oxLDL PAI1	endothelial cells arteries arterioles

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
