# Peer review of "The Pivotal Role of Oxidative Stress in the Pathophysiology of Cardiovascular-Renal Remodeling in Kidney Disease"

_antioxidants, 2021, doi:10.3390/antiox10071041_

Round 1

Reviewer 1 Report

1/ The selection of references for the introduction section appears to be somehow biased. Ref 6-9 refer only to studies on surrogate endpoints using olmesartan. This may look inappropriate since ROADMAP, a placebo controlled trial showed adverse effects of olmesartan on hard cardiovascular endpoints (see NEJM 2011,364,907).

2/ Blockade of the renin-angiotensin system (RAS) do not improve cardiovascular outcome in non-selected hypertensive populations (see JAMA 2002,288,2981). In fact, RAS blockade was proven effective only in patients with heart failure with reduced ejection fraction and chronic kidney disease with heavy proteinuria.

3/ Authors should acknowledge that anti-oxidant therapies have not yet shown a clinically significant benefit in randomized controlled trial.

4/ Authors should now use the consensus nomenclature for kidney disease. "Kidney failure with or without replacement therapy" should be preferred to "end stage renal disease and dialysis" (see Kidney Int 2020,97,1117). Best regards.

Author Response

We thank Reviewer 1 for his/her thoughtful and constructive suggestions. We carefully revised the manuscript according to the comments and have added the additional issues that were raised in the text as requested.

Comment #1: The selection of references for the introduction section appears to be somehow biased. Ref 6-9 refer only to studies on surrogate endpoints using olmesartan. This may look inappropriate since ROADMAP, a placebo-controlled trial showed adverse effects of olmesartan on hard cardiovascular endpoints (see NEJM 2011,364,907).

Response: The Reviewer rightly pinpointed that we referred to specific studies on Olmesartan, however our aim was to underlie the pleiotropic effects of RAS inhibitors (both ARBs and ACEis) that might have a positive influence in the cardiovascular and renal remodeling. We have however replaced those references and have included one updated revision of clinical practice to prevent cardiovascular outcomes involving the use of ARBs and ACEis provided by the European Society of Cardiology (doi 10.1093/med/9780199656653.003.0019_update_001)

Comment #2: Blockade of the renin-angiotensin system (RAS) do not improve cardiovascular outcome in non-selected hypertensive populations (see JAMA 2002,288,2981). In fact, RAS blockade was proven effective only in patients with heart failure with reduced ejection fraction and chronic kidney disease with heavy proteinuria.

Response: We thank the Reviewer for her/his comment. The aim of this review was to discuss the role of oxidative stress in the renal pathophysiology with an outlook over cardiovascular complications which are the most important causes of excessive morbidity and mortality in chronic kidney disease patients. The Reviewer rightly reports that blockade of RAS with ACE inhibitors in non-selected hypertensive did not ameliorate prognosis when compared to calcium channels blockers or thiazides, however, in the study he/she reported (JAMA 2002,288,2981) none of the ARBs was considered at that time. On the other hand, the critical role of the RAS has been well established since decades and it cannot be overlooked that its downregulation serves as renal and cardiovascular protection as has been shown in several studies including the well-known RENAAL, IDNT, IRMA2, DETAIL trials (European Heart Journal Supplements 2009;11:3-8, N Engl J Med 2004;351:1952-61, correction N Engl J Med 2005;352:1731, N Engl J Med. 2001; 345: 851– 860, Diabetes 2006;55:3550-5)

Comment #3: Authors should acknowledge that anti-oxidant therapies have not yet shown a clinically significant benefit in randomized controlled trial.

Response: We agree with the Reviewer that it should be acknowledged that many trials involving antioxidants therapies failed to demonstrate clinically significant benefit, therefore we revised the text accordingly at line 323 (section 5 - Cardiovascular risk factors in CKD and ESRD patients) by providing the relative references which explore literature evidence and the possible explanation for those results (JAMA  2007;297:842-57 and Am J Cardiol 2008;101:14D-19D)

Comment #4: Authors should now use the consensus nomenclature for kidney disease. "Kidney failure with or without replacement therapy" should be preferred to "end stage renal disease and dialysis" (see Kidney Int 2020,97,1117). Best regards.

Response: We modified the text as suggested.

Reviewer 2 Report

This review manuscript discussed the harmful role of RAS and oxidative stress and those mechanisms in heart and kidney in the kidney diseases including ESRD.

  1. In the later part of the paragraph starting at line 187 in page 5 (Oxidative stress in CKD and …), the influences of hypertension and BP variability were discussed. However, the role of oxidative stress was not clearly mentioned.
  2. Table 1 is confusing and difficult to understand. Please add the explanation in the legends more detail. Does the “Compartment” mean the cells in which “Principal effectors” act to cause the “Molecular mechanism”? What is the interstitial cells which seemed not discussed in the body text. In the column of principal effectors, both molecules mediating oxidative stress directly and others seemed to be shown together, therefore they should be separately shown.
  3. At the line of 51 in page 2, juxaglomerular may be mistyping.
  4. The section 5 is too long so please summarized using table or figure, for instance. I think the main theme of this review manuscript will be oxidative stress and its role in the kidney disease. Therefore, you should re-consider the degree of detail in the description about Rho A/ROCK, MYPT-1, and connexins although ROS may affect these molecules.
  5. In the section 6, oxidative stress in hemodialysis and hemodiafiltration was well discussed. Please discuss oxidative stress in peritoneal dialysis more detail.
  6. From the point of view of oxidative stress, please propose the promising therapeutic options other than green tea to prevent kidney disease and/or complications such as cardiovascular diseases.

Author Response

We thank Reviewer 2 for his/her thoughtful and constructive suggestions and the manuscript has been carefully revised according to his/her comments. 

Comment #1: In the later part of the paragraph starting at line 187 in page 5 (Oxidative stress in CKD and …), the influences of hypertension and BP variability were discussed. However, the role of oxidative stress was not clearly mentioned.

Response: Taking into account the Reviewer’s comment we modified the text and have added two references reporting a correlation between blood pressure variability and oxidative stress established by experimental evidences (J Hum Hypertens 2017;31:70–75 and Diabetol Metab Syndr 2019;11:11-29).

Comment #2: Table 1 is confusing and difficult to understand. Please add the explanation in the legends more detail. Does the “Compartment” mean the cells in which “Principal effectors” act to cause the “Molecular mechanism”? What is the interstitial cells which seemed not discussed in the body text. In the column of principal effectors, both molecules mediating oxidative stress directly and others seemed to be shown together, therefore they should be separately shown.

Response: As rightly interpreted by the Reviewer the Table is intended to describe the molecular mechanisms taking part into the cardiovascular-renal remodeling, the principal molecules (effectors) involved in the progression of those processes and the cells in which they take place. However, we have added further explanations in the legend and the term interstitial cells has been removed to avoid misleading interpretations. In addition we included in the text a brief explanation of the interstitial fibrosis process associated to tubular injury.

Comment #3: At the line of 51 in page 2, juxaglomerular may be mistyping.

Response: The mistyping has been corrected.

Comment #4: The section 5 is too long so please summarized using table or figure, for instance. I think the main theme of this review manuscript will be oxidative stress and its role in the kidney disease. Therefore, you should re-consider the degree of detail in the description about Rho A/ROCK, MYPT-1, and connexins although ROS may affect these molecules.

Response: The relationship between Rho A/ROCK pathway, oxidative stress and oxidative stress mediated processes involved in cardiovascular and renal remodeling is highly established and we feel that in a review where the role of oxidative stress mediated cardiovascular-renal remodeling is illustrated, a deeper description of Rho A/ROCK pathway should be considered. However, following the reviewer’s suggestion a figure (Figure 2) summarizing the considered processes has been added. 

Comment #5: In the section 6, oxidative stress in hemodialysis and hemodiafiltration was well discussed. Please discuss oxidative stress in peritoneal dialysis more detail.

Response: We have expanded section 6 including data from our very recent experimental evidence for the role of oxidative stress in peritoneal dialysis (Artif Organs. 2021 May 26. doi: 10.1111/aor.14001. Epub ahead of print, which has been added to the reference list), that at the time of the original submission of our paper could not be included being the relative manuscript not yet published.

Comment #6: From the point of view of oxidative stress, please propose the promising therapeutic options other than green tea to prevent kidney disease and/or complications such as cardiovascular diseases.

Response: We have added in section 5 some therapeutic options such as inhibitors of sodium/glucose cotransporter 2 (SGLT2 inhibitors) and agonists of Glucagon like peptide-1 Receptor (GLP-1R agonists) that have been shown to have important effects on oxidative stress and therefore important role in cardio-nephroprotection. In addition, throughout the manuscript the effects of ROCK inhibition for the reduction of oxidative stress and therefore for cardio-nephroprotection have been reported.

Round 2

Reviewer 1 Report

None

Reviewer 2 Report

None.